# Efficacy of Broilers as a Method of Face Fly (*Musca autumnalis* De Geer) Larva Control for Organic Dairy Production

**DOI:** 10.3390/ani10122429

**Published:** 2020-12-18

**Authors:** Hannah N. Phillips, Roger D. Moon, Ulrike S. Sorge, Bradley J. Heins

**Affiliations:** 1Department of Animal Science, University of Minnesota, Saint Paul, MN 55108, USA; 2Department of Entomology, University of Minnesota, Saint Paul, MN 55108, USA; rdmoon@umn.edu; 3Department of Veterinary Population Medicine, Saint Paul, MN 55108, USA; ulrike.sorge@tgd-bayern.de; 4Bavarian Animal Health Services, 85586 Poing, Germany

**Keywords:** organic, animal welfare, broiler chickens, dairy cattle, pest management, behavior, pasture, face fly, dung fly, fly control

## Abstract

**Simple Summary:**

Dairy cows housed on pasture are commonly afflicted by face flies, which develop as larvae in cow manure. Some organic dairy producers graze chickens in succession behind cattle as an effort to disrupt the development of dung fly larvae, but this method has yet to be validated as a successful method of pest management. In a controlled research trial, pastured broiler chickens did not reduce the survival of face fly larvae that developed in cow dung pats. This finding was buoyed by broiler behavioral observations, which revealed the broilers were unwilling to forage in dung pats. In general, broilers were reluctant to leave their shelter as solar radiation increased and climatic conditions were associated with the displacement of time allocated to sitting and sleeping behaviors. Dairy producers should take heed when adopting methods of fly control that rely on broiler chickens to willingly forage in cow dung pats especially when climatic conditions are aversive.

**Abstract:**

The objective of this study was to evaluate Freedom-Ranger broiler chickens as a method to control face fly (*Musca autumnalis* De Geer) larvae in cow dung pats on pasture. Ninety-nine pats in three replicates were inoculated with first-instar larvae and exposed to one of four treatment conditions for 3 to 4 days: (1) an environment-controlled greenhouse (GH); (2) pasture without broilers (NEG); (3) pasture with 25 broilers stocked at a low density of 2.5 m^2^ of outdoor area per broiler (LOW); and (4) pasture with 25 broilers stocked at a high density of 0.5 m^2^ of outdoor area per broiler (HIGH). Broiler behaviors and weather conditions were recorded twice daily. Survival rates of larvae (mean, 95% CI) were similar for pats in the NEG (4.4%, 2–9%), LOW (5.6%, 3–11%), and HIGH (3.2%, 2–7%) groups, and was greatest for larvae reared in the GH (54.4%, 36–72%) group compared to all other groups. The proportion of broilers observed pasture ranging was 14.0% (6–28%) but was negatively related to solar radiation. Broilers were never observed foraging in pats. Results indicate that use of broilers may not be an effective method for controlling larvae of dung pat breeding flies.

## 1. Introduction

On organic dairy farms in the United States of America (USA) and Europe, the face fly (*Musca autumnalis* De Geer) is a common pest of pastured cattle. Adults of the face fly feed primarily on excretions around their host’s eyes, and are notorious vectors for the bacterium *Moraxella bovis*, which causes infectious bovine keratoconjunctivitis—commonly known as pinkeye [1]. Cattle attacked by face flies may cope by bunching in a group with heads toward the center [2], and increasing the rate of head throws as the number of face flies increase [3]. Because face flies harm cattle, suppressing numbers of this pest may improve cattle welfare.

The use of synthetic substances for fly control on organic cattle is restricted by regulations set forth by the United States Department of Agriculture National Organic Program, which maintains official federal standards for organic production practices (§205.603) [4]. The inclusion of multiple effective fly control methods (i.e., integrated pest management) is an important approach for managing dung breeding flies on organic dairy farms. For example, plant-derived topical products may repel horn flies (*Haematobia irritans* L.) on cattle for up to 1–3 days after application [5,6], and modern walk-through systems that trap to kill adult horn flies may reduce the number of horn flies on cattle by 44–75% [7,8]. Some previous studies suggest that face flies may also be repelled with plant-derived repellents [9]; yet, effective methods for the control of face flies on organic cattle are not well documented.

Source reduction of dung breeding flies with free ranging chickens may be a feasible option for organic dairy farms. Adult female dung flies lay their eggs in fresh cow manure, which provides nutrition and safety for the development of immatures [10]. Once the larvae mature, they burrow under the dung pat to pupate and finish developing into adult flies. The development of an egg into a pupa can take as little as 7 days in the heat of summer, and to adult takes 12–28 days [10]. Strategies that biologically disrupt the developing larvae could ultimately reduce the proportion of larvae that reach adulthood and may represent a feasible method of pest management on organic dairy farms.

Some producers believe that chickens will consume dung fly larvae in dung pats, and that grazing cattle and chickens in succession as part of a diversified system is an effective method of disrupting dung fly developments [11]. In a survey conducted in 2012, Sorge et al. [12] reported that 9% of organic dairy farms in Minnesota used foraging chickens as a method of controlling dung flies. In a survey of 18 farmers in California, 17% reported a pest control benefit after adopting pastured poultry practices [13]. However, using chickens as a method of controlling dung flies has yet to be evaluated under experimental conditions. The high protein content and digestibility of fly larvae make them a potentially excellent addition to the chicken diet [14], and previous research indicates that the diet of chickens with access to pasture may consist of up to 9% insects on a dry matter basis [15]. However, no scientific studies have determined whether chickens can successfully reduce the survival rate of dung fly larvae in cow dung pats on pasture, making this only an anecdotal method for effective fly control so far.

Alternative management strategies used in organic livestock production require support from controlled research trials to confirm that the strategies indeed improve animal welfare. Research surveying organic dairy producers and veterinarians in the USA acknowledged that the lack of scientific support for certain practices utilized by organic livestock producers jeopardizes animal welfare [16,17]. Flies are considered an important animal welfare concern due to the negative effects flies may have on the affective state, behavior, and health of organic dairy cattle [18]. Furthermore, ineffective management of fly pests may further impair animal welfare, and therefore represents a major threat to organic dairy animal welfare. In a review of dairy industry changes that affect animal welfare, Barkema et al. [19] suggested that future research should begin classifying effective and ineffective organic-approved management strategies.

Little is known about broiler behaviors, such as movement on pasture, time budgets, and foraging from dung pats or elsewhere. The consumption of pasture contents, including forages, insects, and larvae, may be affected by weather and stocking density [20]. Therefore, it is critical to assess behaviors and factors affecting behaviors to support findings of studies that depend on consumption of pasture contents.

The objectives of this study were: (1) to determine if broiler chickens affect the survival rate of face fly larvae presented in cow manure pats on pasture, and (2) to assess broiler pasture ranging and behaviors and their responses to weather conditions.

## 2. Materials and Methods

### 2.1. Animal Care and Housing

The University of Minnesota Institutional Animal Care and Use Committee approved all animal care and procedures specific to this experiment (protocol number #1607-33960A).

The experiment was conducted from June to August 2018 at the West Central Research and Outreach Center (Morris, MN, USA) in pastures that were consecutively grazed by lactating dairy cows (*Bos taurus* L.) and broiler chickens (*Gallus gallus domesticus* L.). The dairy herd and pastureland used in this study had been certified organic since 2010 by the Midwest Organic Services Association, following regulations set forth by the United States Department of Agriculture National Organic Program. Cows in this study were housed on pasture for 22 h per day and spent the remaining 2 h per day for milking procedures which took place twice daily in a swing-nine para-bone milking parlor at 06:00 h and 17:00 h. A total of 80 cows grazed the pastures used in this study, which were rotationally stocked at a rate of 4 cows per ha. Cows rotated to a new paddock every 2 days based on forage biomass availability. Pastures included perennial forbs, grasses and legumes, such as alfalfa (*Medicago sativa* L.), chicory (*Cichorium intybus* L.), meadow brome grass (*Bromus riparius Rehmann*), meadow fescue (*Schedonorus pratensis (Huds.) P. Beauv*), orchard grass (*Dactylis glomerata* L.), perennial ryegrass (*Lolium perenne* L.), red clover (*Trifolium pratense* L.) and white clover (*Trifolium repens* L.). Cows had ad libitum access to minerals and water, and were supplemented with 2.72 kg of organic corn grain daily.

The study involved a total of 150 mixed-sex Freedom Ranger broiler chickens (Welp Hatchery, Bancroft, IA, USA) in three replicate groups of 50 that hatched on 1 May, 29 May, and 9 July, respectively. The Freedom Ranger hybrid consisted of a four-line cross developed in the 1960’s to meet the French Label Rouge Free Range program standards. For each replicate group, day-old broiler chicks that were vaccinated for Marek’s at the hatchery were housed in a 2.22 m^2^ pen bedded with 5 cm of wood shavings and had ad libitum access to feed and water. Broilers in each group began acclimating to experimental housing at 4 weeks of age and were randomly assigned to pens with 2.5 m^2^ or 0.5 m^2^ of adjacent pasture per bird. These stocking densities were chosen based on animal welfare standards for “Free-Range” and “Pasture-Raised” chickens [21,22]. Pens were balanced by bird sex and weight and housed 25 broilers in each half of a floorless 3.7 × 3.7 m mobile shelter (Chicken Ranger Coops, Narvon, PA, USA) that was divided into two equally sized areas (1.8 × 3.7 m). One door (0.91 × 0.91 m) per pen allowed broilers free choice between the shelter and pasture during the day, and confinement to the shelter at night to protect the birds from predators. The perimeters of the pens were fenced with 1.2-m tall portable electric poultry netting (PoultryNet, Premier1Supplies, Washington, IA, USA) that was continuously charged by a solar powered 0.60 joule energizer (IntelliShock, Premier1Supplies, Washington, IA, USA).

For the duration of the study, each group of broilers had ad libitum access to water and granite grit, and were fed a restricted diet of 113 g of concentrate per bird (20% crude protein; Chick Starter AMP, Vita Plus Corporation, Madison, WI, USA) daily at dusk (between 21:00 h and 22:00 h) prior to shelter confinement and removed between 05:00 h and 06:00 h the next morning to encourage pasture foraging. The health of each bird was assessed prior to study initiation, starting at 4 weeks of age and weekly thereafter. Hock lesions and foot pad lesions were assessed during each weekly heath assessment. No broilers had hock or foot pad lesions or showed signs of gait difficulties.

### 2.2. Experimental Design

This study was conducted as a randomized complete block design with repeated sampling, blocked by replicate group. On each block’s day of treatment initiation (21 June, 24 July, and 15 August), fresh dung was collected between 05:00 h and 06:00 h from randomly selected cows in neighboring paddocks, homogenized by hand mixing, and stored in a covered bucket until use the same morning. Inocula for dung pats were prepared by counting and transferring aliquots of 100 first-instar face fly (*M. autumnalis*) larvae into 5 g of dung in each of 33 covered Petri dishes. Larvae were obtained from a lab colony. Petri dishes and their contents were stored at 23 °C to delay development until treatment assignment. At 12:00 h, one-liter dung pats (33 per replicate group; 99 total) were consecutively deposited in treatment conditions and inoculated with maggots by random assignment among the four treatment conditions: (1) on 8 cm of sand in 5-L buckets in a naturally ventilated environment-controlled greenhouse (GH, *n =* 9 pats); (2) on pasture without broilers (NEG, *n* = 30 pats); (3) on pasture with broilers stocked at a low density (LOW, *n =* 30 pats); or (4) on pasture with broilers stocked at a high density (HIGH; *n* = 30 pats). 

Dung pats were placed equal distances apart in the outdoor portion of the broiler pens for the LOW (10 pats 0.9 m apart) and HIGH (10 pats 0.5 m apart) treatment groups, and adjacent to the broiler pens for the NEG treatment group (10 pats 0.9 m apart). Pats were inoculated by each Petri dish into the center of a recipient pat. Treatment within replicate served as the experimental unit (3 replicates of 4 treatments per rep = 12) and pat within experimental unit was treated as a sub-sample. Over the three replicates, average (± SD) broiler age was 48 ± 10 days and average broiler weight was 1.94 ± 0.7 kg at the onset of the experiment.

### 2.3. Data Collection

Once inspections indicated the transplanted larvae reached the third-instar, after 3–4 days, pats and the 3 cm of soil underneath pats in the NEG, LOW, and HIGH treatment groups were transferred to 5-L buckets with 8 cm of sand. The buckets were housed in the greenhouse neighboring the buckets housing pats of the GH treatment group until the larvae began to pupate after 3–5 days. The number of larvae and pupae were then counted in each bucket by wet sieving sand and pats through a 1.41 × 1.41 mm square wire mesh sieve to extract surviving larvae and pupae.

Behavior observations were recorded by an observer for the second and third replicates of the study in the morning (between 09:00 h and 11:00 h) and afternoon (between 13:00 h and 17:00 h) when precipitation was not expected for a total of two observation periods per day. Prior to behavior observations, pasture ranging was recorded for each pen as the proportion of broilers outside of the shelter. Behaviors were then recorded in continuous 60 s observation periods on 10 individual focal broilers per pen using the Animal Behaviour Pro mobile app (version 1.2) [23]. Focal broilers were identified using livestock paint and were observed in random order alternating between treatment pens. Behavioral states corresponding to the time budget were recorded as durations and foraging events were recorded as binary outcomes (i.e., the occurrence of foraging behaviors within the 60 s observation period was recorded as either a yes [presence] or no [absence]). The frequency of foraging bouts was not recorded. An ethogram defining recorded behaviors is in Table 1.

The University of Minnesota West Central Research and Outreach Center weather station recorded measures of ambient humidity, ambient temperature, precipitation, solar radiation, and wind speed every 15 min. The comprehensive climate index (CCI) was calculated to describe the apparent temperature based on ambient humidity, ambient temperature, solar radiation, and wind speed variables [25]. For each pasture ranging and behavioral observation, the time was rounded to the nearest 15 min interval and matched with the climatic condition data.

### 2.4. Statistical Analysis

Logistic regression models with beta error distributions and logit link functions were used to analyze seven binomial outcomes for larval survival, and broiler pasture ranging and behaviors. Modeling was accomplished in RStudio (version 1.3.1073) [26] using the *glmmTMB* package [27]. All models included fixed effects of treatment and replicate, and a random effect of treatment within replicate to account for the dependency among repeated sampling within experimental units.

Nonparametric correlation coefficients, Spearman’s rho, were used to examine pair-wise relationships between pasture ranging and behaviors and weather conditions at each time of observation. Stocking density treatment could not be included as an independent variable in correlations, so each correlation was initially performed on each stocking density treatment and replicate; there were no differences in direction or *p*-values between stocking density. Therefore, observations were pooled, and the correlation coefficient is reported with the corresponding degrees of freedom (df).

The one weather variable with the strongest correlation with the chosen dependent variable was included in the models for pasture ranging and behaviors. For the analysis of pasture ranging, a fixed covariate of solar radiation was included. The analyses for behaviors of the time budget included a fixed covariate of CCI and a random effect of broiler ID to account for repeated sampling. For the analysis of foraging events, pen averages were taken to represent the proportion of broilers observed performing the behavioral event for each observation and no covariate was included in the model based on the absence of a relationship between foraging and climatic conditions.

Type II Wald χ^2^ tests were used to test the significance of main effects and are reported with the corresponding degrees of freedom followed by number of observations. The Tukey adjustment was applied to compare groups when the corresponding main effect had *p* ≤ 0.05. Marginal mean rates and 95% confidence intervals (CIs) for all responses were transformed to the natural scale. Treatment groups were compared using rate ratios (RRs), the ratio of mean rates between two groups.

## 3. Results

### 3.1. Climatic Condition and Precipitation

Daily weather conditions over the course of the experiment are displayed in Figure 1. Study replicates occurred from 21–25 June (replicate 1), 24–27 July (replicate 2), and 15–19 August (replicate 3). The average ambient temperature recorded during study replicates 1, 2, and 3 were 22.3 °C, 19.0 °C, and 21.4 °C, respectively. The ambient temperature range (minimum–maximum) recorded during study replicates 1, 2, and 3 were 15.0–28.9 °C, 11.7–28.3 °C, and 15.0–28.3 °C, respectively. Total precipitations accumulated during study replicates 1, 2, and 3 were 7.1 mm, 9.7 mm, and 0.0 mm, respectively. Precipitation during study replicates 1 and 2 took place on 24 June at 02:45–03:00 h and on 25 July at 04:00–07:00 h, respectively.

### 3.2. Face Fly Larva Survial

Larval survival rates by treatment are shown in Figure 2. Upon transferring pats from the pasture, evidence of trampling but not scratching nor pecking was apparent for almost every pat in the broiler treatment groups (LOW and HIGH), but not for the pasture control group without broilers (NEG). There was an effect of treatment on the survival rate of larvae (χ^2^_(df=3, n=98)_ = 55.2, *p* < 0.01).

For the effect of treatment, the survival rates (95% CI) were 54.4% (36–72%), 4.4% (2–9%), 5.6% (3–11%), and 3.2% (2–7%) for pats in the GH, NEG, LOW, and HIGH groups, respectively. The survival rate for larvae in the GH treatment group was greater (RR, 95% CI) compared to the NEG (11.9, 4–34), LOW (9.4, 3–26) and HIGH (16.3, 5–49) treatment groups (*p* < 0.01). The NEG, LOW, and HIGH treatment groups had similar survival rates (*p >* 0.70).

### 3.3. Broiler Behaviors

The means (range) for CCI, ambient humidity, ambient temperature, solar radiation, and wind speed recorded during observations were 30.7 °C (15–40 °C), 73% (57–87%), 23.2 °C (16–28 °C), 415 W/m^2^ (64–679 W/m^2^), and 0.74 m/s (0–1.8 m/s), respectively.

#### 3.3.1. Pasture Ranging

There was an effect of solar radiation on pasture ranging, such that ranging decreased as solar radiation increased (χ^2^_(df=1, n=22)_ = 8.9, *p* < 0.01). There was no effect of stocking density on pasture ranging. On average, only a small proportion of the flock was observed pasture ranging (mean = 14.0%, 95% CI = 6–28%). Furthermore, no birds were observed pasture ranging for over a third (36%) of the observations. Results for the analysis of pasture ranging indicate that broilers are less likely to range in the pasture during periods of high solar radiation regardless of stocking density treatment and suggest that overall pasture use is low for the experimental conditions of the study, which were characterized by open cattle grazing pasture and lack of tree cover.

Correlations between pasture ranging and other climatic conditions recorded during observations are shown in Table 2. The correlations indicate a negative relationship between pasture ranging and CCI and ambient temperature, and a positive relationship between pasture ranging and ambient humidity. Excessive heat and solar intensity appear to be important influencers of pasture ranging in broilers. These results suggest that the hybrid broilers used in this study under these experimental conditions characterized by no tree cover avoid overheating by reducing activity in sun and seek shade as levels of heat stress increase.

#### 3.3.2. Time Budget

The time budget estimates for the effect of CCI are visualized in Figure 3. For the analyses of time budget behaviors, there was no effect of stocking density treatment on any time budget behavior. The effect of CCI only existed for the analyses of standing (χ^2^_(df=1, n=199)_ = 6.2, *p =* 0.01) and sleeping (χ^2^_(df=1, n=199)_ = 4.6, *p =* 0.03). On average, the time budget (95% CI) was comprised of 41.4% (31–52%) sitting, 30.1% (20–43%) standing, 14.1% (8–24%) sleeping, and 1.4% (1–3%) traveling.

The time budget analyses indicated that for every 1 °C increase in CCI, the proportion of time broilers were observed standing increased by a factor of 1.10 (regression coefficient = 0.10, 95% CI = 0.02–0.17). Alternatively, the proportion of time broilers were observed sleeping decreased by a factor of 0.92 for every 1 °C increase in CCI (regression coefficient = −0.09, 95% CI = −0.17–−0.01). These results indicate that increased CCI values between 16 °C and 39 °C may disrupt sleeping and standing behaviors.

Correlations between time budget behaviors and other climatic conditions recorded during observations are shown in Table 2. In general, all correlations indicated weak strengths of association. Sitting had positive relationships with ambient temperature and solar radiation, whereas sleeping had negative relationships with the same noted climatic conditions. Sleeping also had a positive association with ambient humidity. Traveling had a negative association with wind speed. Sleeping appeared to be the most affected by climatic conditions based on the correlation coefficients.

#### 3.3.3. Foraging

For the analysis of ground foraging, there was no effect of stocking density treatment. On average, the percentage of broilers recorded ground foraging during observations was 21.5% (95% CI = 6–54%). At least one broiler ground foraged during 90% of observations. Broilers were never observed pat foraging, but predation of insects and frogs was observed during ground foraging events.

Correlations between ground foraging and climatic conditions recorded during observations indicated that ground foraging was not associated with any climatic condition (Table 2).

## 4. Discussion

Freedom Ranger broilers had no effect on survival rates of face fly larvae. This study is the first to specifically examine predation on dung fly larvae by pastured poultry. Related studies on the utilization of poultry as fly predators reported that pastured laying hens were successful at managing weed seeds [28]. Furthermore, Muscovy ducks (*Cairina moschata* L.) successfully consumed and reduced populations of house fly (*Musca domestica* L.) larvae and adults in closed calf rooms, but not in open maternity pens [29,30]. Another study showed that six-week old Barred Plymouth Rock chickens (*Gallus gallus domesticus*) and African geese (*Anser cygnoides* L.) reduced the number of insect pests in an apple orchard intercropped with potato crops [31]. Under further investigation, this study also reported that muscid (Muscidae) larvae and adults were found in 0% and 33 to 40% of chicken crops, respectively (ibid). A possible limitation of the current study is that transferring pats from the pasture to the greenhouse prior to larva pupation may have limited the potential for broilers to prey on pupae, but there were few larva survivors anyway, so this artifact was likely small. Results from previous research [29,30,31] agrees with the results of the current study, which indicate that broilers may not be successful predators of dung fly larvae that pass their developmental stages in cattle manure.

It is unknown whether face fly larvae are truly an attractive feed source for broilers. Anecdotal evidence suggests that face fly larvae and pupae are highly sought after when they are hand-fed to broilers. In an early study, Dashefsky et al. [32] successfully fed face fly pupae to day-old White Rock chickens, yet the level of palpability was not determined or discussed. Other studies [33,34,35,36] suggested that black soldier fly (*Hermethia illucens* L.), house fly (*Musca domestica* L.), and yellow mealworm (*Tenebrio molitor* L.) larvae can successfully be used as a palatable feed supplement for poultry. Although research on the palatability of face fly larvae for poultry is sparse, insect larvae in general is a highly attractive feed source for poultry.

The Freedom Ranger broilers used in the current study were a hybrid cross of several strains and were originally bred for their suitability for pasture housing systems. Previous studies [37,38] suggest that hybrid strains that demonstrate slower growth and improved mobility compared to pure strains make hybrids more equipped for free-range systems. A study by Lambertz et al. [37] demonstrated that hybrid male broilers (Bresse-Gauloise × New Hampshire) grew faster but had similar carcass quality to their purebred equivalent (Bresse-Gauloise). Other slow-growing hybrid broilers (Labresse × L86) were previously demonstrated to have increased pasture ranging, improved gait scores, and reduced dermal lesions compared to their fast-growing counterpart (Ross 208) [38]. Alternatively, another study [39] reported that a slow-growing pure strain (White Bresse L40) demonstrated increased pasture ranging and foraging behaviors compared to a fast-growing hybrid strain (Kosmos 8 Red), suggesting that speed of growth may be an important influencer of broiler behavior.

Stark differences in larvae survival rates were apparent between pats reared in the pasture treatment groups (NEG, LOW, and HIGH) and those raised in the greenhouse (GH). Valiela [40] also reported lower face fly larvae survival rates when reared in field conditions compared to laboratory conditions for the first 5 days after hatching, in which the majority of morality occurred on the first day of field exposure. Results from the current study are similar to those of the experiments conducted in this previous study (ibid), which indicate that pasture conditions dramatically increase the mortality of face fly larvae.

The overall survival rates of larvae in this study were lower than previous reports. Valiela [40] reported greater mean survival rates of 78–76% and 42–31% after 3 to 5 days of exposure to laboratory conditions maintained at a constant temperature of 32 °C and field conditions, respectively. The reduced survival of face fly larvae of the current study may have been caused by several factors, including temperature and predation.

Cool temperatures may have increased the mortality of face fly larvae for the current study. First-instar larvae are especially prone to temperature extremes since their mobile abilities are not fully developed. In an experiment on the survival of face fly first-instar larvae, Valiela [40] found that constant temperatures of 35 to 40 °C maximized the survival rate at 69%, whereas temperatures of 10 °C, 16 °C, and 20 °C yielded survival rates of 0%, 22%, and 53%, respectively. This study reported mortality rates of 1 to 2% per hour at constant temperatures from 10 to 20 °C (ibid). Within the first 24 h of pat placement for the current study, temperatures of ≤16 °C were not observed; however, temperatures of ≤20 °C were observed in single durations of 8 h, 10 h, and 8 h for replicates 1, 2, and 3 of the study, respectively. Therefore, low temperatures may have accounted for 8 to 20% of face fly larvae mortality in pasture treatment groups. Alternatively, the greenhouse environment may have provided a more temperate environment conducive for better early life face fly larvae survival, as it was protected from extreme temperature fluctuations. Yet, the survival rate of face fly larvae reared in the greenhouse was lower than reported in previous studies where larvae were reared in cow pats under laboratory conditions. Although the greenhouse may have somewhat shielded pats from extreme temperature fluctuations, it is still possible that temperatures dropped below 20 °C, resulting in larvae mortality. Therefore, cool temperatures observed during the current study may have reduced the survival of face fly larvae in all treatment groups.

Naturally occurring predatory arthropods may have consumed some of the face fly larvae. It is possible that dung beetle (*Sphaeridium* spp.) larvae and rove beetles (*Philonthus cruentatus* Gmelin) preyed upon face fly larvae [41,42]. A study conducted at a prairie cattle pasture confirmed the presence of dung beetles (*S. lunatum* Fabricius and *S. scarabaeoid* L.) and rove beetles (*P. cruentatus*) in Minnesota [43]. For the current study, there was evidence of beetle tunneling for every undisturbed pat in the pasture treatment groups, suggesting the presence of predatory beetles. In a study to investigate the arthropod predation on face fly larvae, Valiela [40] found that the introduction of dung beetles (*S. scarabaeoid*) and rove beetles (*P. cruentatus*) reduced the larvae survival rates by 14% and 57%, respectively. Moreover, the presence of both predators reduced the larvae survival rate by 66% (ibid). Based on this information, it is possible that naturally occurring dung beetles and rove beetles preyed on face fly larvae of the current study, which would explain the majority of the survival differences among pats raised in the greenhouse and pats raised in the pasture.

It is possible that the cattle dung use in this study reduced the survival of fly larvae due to the presence of alkaloids. A recent study by Parra et al. [44] reported that the dung from cattle that consumed endophyte-infected fescue contained two important alkaloids, peramine and lolitrem B, which reduced horn fly (*H. irritans*) larval survival from 45% to 10%. Based on this information, it is possible that the consumption of perennial ryegrass (*L. perenne*) included in the pastures of the current study resulted in the presence of these alkaloids and consequently lowered the survival of the fly larvae.

A fundamental necessity for the successful utilization of broilers as a method of controlling dung fly immature development is pasture ranging. For the current study, only an average of 14.0% of broilers were observed ranging, which was negatively affected by increasing solar radiation. Previous studies using slow-growing strains (Delaware, Labresse × L86, and Sherwood White) similarly reported that 9 to 25% of broilers were observed ranging [38,45,46] and that 34 to 38% of observations resulted in lack of ranging [46]. Stadig et al. [47] similarly reported that increased solar radiation had a negative effect on slow-growing broiler (Sasso T451) ranging for values of 0 to 1000 W/m^2^. Hegelund et al. [48] reported a negative relationship between laying hen (ISA Brown and Lohmann Brown) pasture ranging and ambient temperatures of 17 to 41 °C, which agrees with correlation results of the current study. The low pasture ranging observed in this study may also be partially explained by an undesirable pasture habitat. It is intuitive that chickens prefer covered areas since their domestication evolved as descendants of Red Jungle Fowl (*Gallus gallus* L.), which rely on vegetative cover for protection from predators [49]. Since the pasture space of the current study was open and comprised of forages that were 9 to 11 cm tall, it was not surprising that broilers of this study were commonly observed inside the shelter where safety from solar radiation and predators could be preserved. It is possible that a more temperate climate with cooler summers would help promote pasture ranging in a similar treeless pasture habitat used for cattle grazing. However, cattle pastures without trees or shelters would still be unsuited for promoting pasture ranging even under optimal weather conditions since they do not provide protection from predators.

Outdoor structural enrichments providing cover may make pasture ranging more desirable to broilers by filtering solar radiation that causes excessive heat [46] and providing a sense of protection from aerial predators [49]. For example, canopy enrichments were previously demonstrated to improve range utilization in areas up to 20 to 31 m from the shelter [46,50]. Dawkins et al. [45] reported that slow-growing broilers (Sherwood White) preferred ranging in areas with tree cover as opposed to areas with short grass in areas 10 m from the shelter. Stadig et al. [47] found that more slow-growing hybrid broilers (Sasso T451) left the shelter and ranged >5 m from the shelter when provided tree cover compared to artificial cover. However, studies conducted by Dawkins et al. [45] and Fanatico et al. [46] similarly concluded that slow-growing broilers (Delaware and Sherwood White) are reluctant to leave their shelter even when offered cover in the pasture area. Although access to pasture may be plentiful, broilers realistically spend the majority of their time in the shelter and away from any potential opportunities to forage for dung fly larvae in cow pats.

Behavioral observations supplement the face fly larvae survival findings of the current study. Broilers were never observed foraging in the dung pats. In fact, broilers spent most of their time sitting. Fanatico et al. [46] also reported that sitting was more commonly observed than standing or walking in 7- and 10-week-old slow-growing Delaware broilers with access to pasture. The time spent standing and sleeping was associated with climatic conditions. Furthermore, only about 22% of broilers were observed performing ground foraging behaviors during observations. In agreement with the current study, Fanatico et al. [46] similarly observed foraging for 28% of observations when averaged across age groups and pen locations for slow-growing Delaware broilers. A possible limitation of this study includes infrequent behavioral sampling. Continuous sampling from dusk until dawn would provide a more accurate estimate of behaviors.

It is unknown whether or not broilers of an age group older than that used in the current study (approximately 7-weeks-old) would have yielded different study results. Fanatico et al. [46] reported that outdoor foraging events increased by a factor of 1.7 between 7 and 10 week-old slow-growing broilers. Almeida et al. [39] similarly reported that outdoor foraging increased between the ages of 11 and 15 weeks for both slow- and fast-growing broiler strains, but acknowledged that broilers rarely consumed larvae or pupae based on an analysis of crop contents. Broilers of the current study were never observed foraging in dung pats for larvae and it seems reasonably unlikely that they would suddenly include this novel foraging technique to their behavioral repertoire as they approach slaughter weight at approximately 12 weeks of age. Yet, it is also possible for broilers to learn specialized foraging strategies depending on the social structure of the flock since domestic fowl engage in social learning during foraging events [51]. Therefore, an interaction between age and social learning of the flock may affect the success of foraging for fly larvae in dung pats for broilers.

To our knowledge, this is the first study to provide evidence that Freedom Ranger broilers do not forage for face fly larvae in cow dung pats in uncovered cattle pasture. Future research should investigate other poultry types and species, including laying hens, ducks, and geese, to fully understand whether poultry may be used as a biological control for managing dung flies on organic dairy farms.

## 5. Conclusions

Pastured Freedom Ranger hybrid broiler chickens stocked at 2.5 m^2^ and 0.5 m^2^ of outdoor area per broiler had no effect on the survival of face fly larvae in cow dung pats in this study. Larva survival rates were greater when reared in an environmentally controlled greenhouse compared to those reared on pasture. Solar radiation had a moderate to strong negative association with broiler pasture ranging. The comprehensive climate index (i.e., apparent temperature) was associated with broiler allocations of time spent sitting and sleeping, indicating that weather conditions may displace broiler time budgets. Broilers were never observed foraging in dung pats but were often observed foraging in other areas of the pen. Broiler pasture ranging and behavioral results indicate that weather conditions may affect the behaviors necessary for dung fly larva predation, but nevertheless pastured Freedom Ranger hybrid broiler chickens were not a successful method of face fly larva control in this study.

## Figures and Tables

**Figure 1 animals-10-02429-f001:**
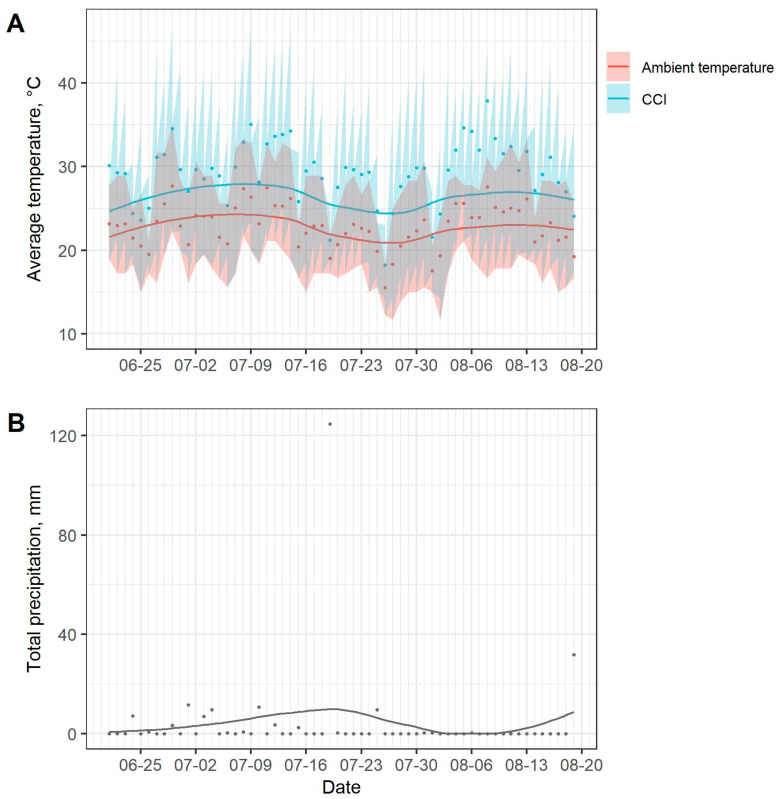
(**A**) Average daily ambient temperature and average daily comprehensive climate index (CCI), apparent temperature. Transparent bands represent daily minimum and maximum values. (**B**) Total daily precipitation.

**Figure 2 animals-10-02429-f002:**
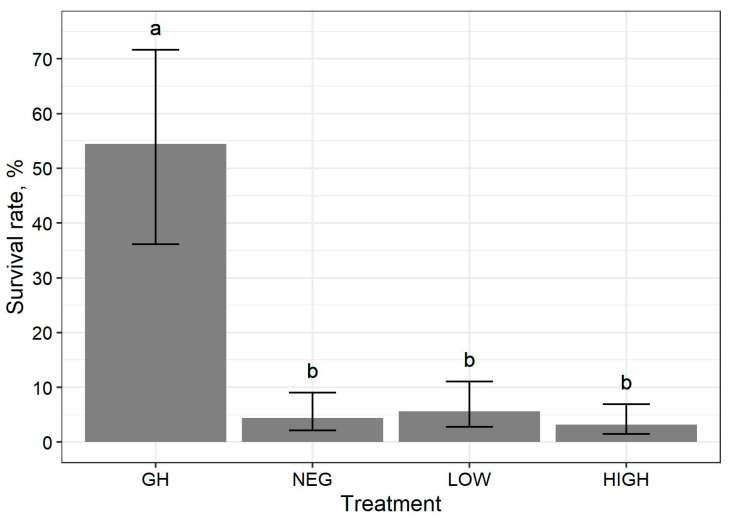
Mean survival rates (± 95% CI) of face fly (*Musca autumnalis* De Geer) larva reared in cow dung pats under different treatment conditions. Treatment means with a different letter are different at *p* ≤ 0.05. Treatments: GH = greenhouse; NEG = on pasture without broilers; LOW = on pasture with low density broilers (2.5 m^2^ outdoor area per bird; HIGH = on pasture with high density broilers (0.5 m^2^ outdoor area per bird).

**Figure 3 animals-10-02429-f003:**
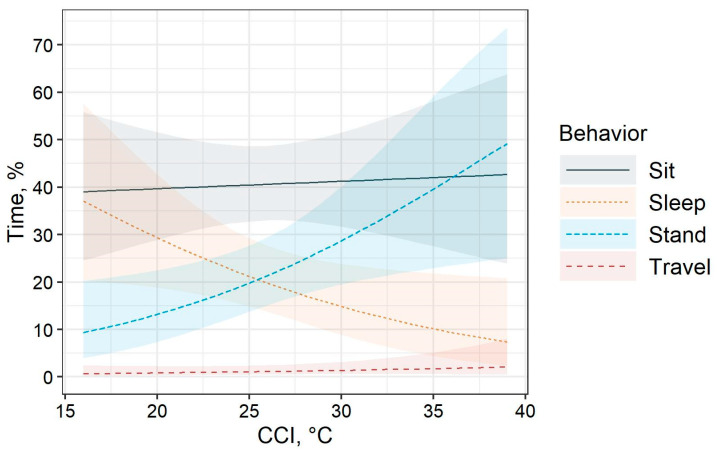
Mean percentages of time (±95% CI) broilers performed time budget behaviors (sitting, sleeping, standing, and traveling) for the effect of CCI. The effect of CCI had *p =* 0.01 and *p =* 0.03 for the analyses of standing and sleeping, respectively. CCI = comprehensive climate index, apparent temperature.

**Table 1 animals-10-02429-t001:** Ethogram of behaviors recorded before and during 60-s observation periods. Modified from Ventura et al. [24].

Behavior	Definition
Pasture ranging	Proportion of flock outside the shelter before start of behavior observation
Time budget ^1^	
Sit	Bird has its breast in contact with the ground. Eyes are open
Stand	Bird maintains upright position on its extended, stationary legs
Sleep	Bird has its breast in contact with the ground. Eyes are closed
Travel	Bird is displaced on the ground, in which the action of legs propels the bird
Foraging ^2^	
Ground foraging	Bird pecks or scratches at the ground
Pat foraging	Bird pecks or scratches at dung pat

^1^ Time budget behaviors are mutually exclusive. Recorded as duration; ^2^ Foraging behaviors are non-mutually exclusive, and foraging behaviors and time budget behaviors are non-mutually exclusive. Recorded as binary outcomes.

**Table 2 animals-10-02429-t002:** Spearman’s rank correlation coefficients and *p*-values for relationships between broiler pasture ranging, time budget behaviors, and ground foraging with climatic conditions.

Behavior (df) ^1^	CCI ^2^	Temperature	Humidity	Solar Radiation	Wind Speed
Pasture ranging (20)	−0.43 (0.05)	−0.54 (0.01)	0.53 (0.01)	−0.68 (<0.01)	−0.13 (0.55)
Time budget (197)					
Sit	0.17 (0.02)	0.18 (<0.01)	−0.10 (0.17)	0.13 (0.06)	−0.05 (0.46)
Stand	0.04 (0.56)	−0.01 (0.94)	−0.10 (0.18)	0.01 (0.87)	−0.01 (0.89)
Sleep	−0.17 (0.02)	−0.14 (0.04)	0.16 (0.02)	−0.14 (0.05)	0.06 (0.42)
Travel	0.13 (0.06)	0.05 (0.47)	−0.03 (0.67)	0.05 (0.72)	−0.20 (<0.01)
Ground foraging (18)	0.16 (0.51)	0.17 (0.47)	−0.31 (0.19)	0.10 (0.69)	0.30 (0.19)

^1^ df = degrees of freedom for Spearman’s rank correlation test; ^2^ CCI = comprehensive climate index, apparent temperature.

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
