# Peer review of "Efficacy of Broilers as a Method of Face Fly (*Musca autumnalis* De Geer) Larva Control for Organic Dairy Production"

_animals, 2020, doi:10.3390/ani10122429_

Round 1
Reviewer 1 Report
The manuscript titled “Efficacy and behaviors of broilers as a method of face fly (Musca autumnalis De Geer) larva control for organic dairy production”, describes a very interesting research on the possible role of broilers in the control of an important bovine pest, face fly, under organic management conditions. This work has a very interesting agricultural-livestock-social component, because the authors search through science (scientific method) to give an explanation to a cultural practice that is used by farmers: control pests through the use of broilers. Overall, the manuscript is very well presented. In general, the methodology is adequate for what it is proposed to evaluate and the experimental design is well designed and established. Although English is not my mother tongue, the manuscript is very well written. The main contribution is that they conclude that there is no real controlling effect of broilers on the M. autumnalis, which is valuable and with them it remains a "belief" what is applied by some producers who indicate that chickens consume dung fly in dung pats being an effective method of disrupting dung fly development.
However, I have some doubts that it would be good for the authors to analyze before publication.
Abstract: I suggest including the objective of the investigation.
Introduction:
In the introduction, reference is made to problems caused by species of the Diptera order to cattle. Among them are named and described the damage caused by two species: M. autumnalis and Haematobia irritans. Their life cycles are described, ending with the need to search for disruption methods to control these pests. However, the criteria for choosing to work with the M. autumnalis and not with H. irritans is not indicated. I suggest removing the information regarding H. irritans or including the criteria to choose only one species.
Results and discussion:
Figure 2 is the most important in the study. With this figure, the authors discuss and conclude that broilers do not significantly affect M. autumnalis survival (HIGH, NEG and LOW) compared to GH treatment. Although the possible factors that could have influenced these results are discussed, there are important points that could have been discussed. For example, the authors in the methodology describe a wide botanical variety in the pastures on which the cows were fed. Among them, there is Lolium perenne which is a forage species characterized by presenting a symbiotic association with chemical compounds of the alkaloid type (peramine, lolitrem B among others). It has been determined that these compounds are transferred from the feeding of the animal through its digestive tract and are eliminated through its dung. In an interesting study, it was determined that these dung had traces of these alkaloids and significantly reduced the survival of H. irritans in laboratory and field conditions (see Parra, L., A. Mutis, M. Chacón, M. Lizama, C. Rojas, A. Catrileo, O. Rubilar, G. Tortella, M.A. Birkett & A. Quiroz. 2016. Horn fly larval survival in cattle dung is reduced by endophyte infection of tall fescue pasture. Pest Management Science 72(7): 1328-1334.). This fact could be part of the discussion in terms of explaining the low survival of larvae from the treatments since they were kept in dung that came from the field.
Lines 299-301. “Results previous research agrees with….” What are those results? References?
I have the feeling that this work focused more on the behavior of chickens in the face of climatic factors than on the effectiveness of broilers as controllers of face fly. Although the title incorporates both themes (efficacy and behavior), the research question of this work is whether or not there is a controlling effect of broilers on the face fly?
Author Response
Thank you so much for taking the time to review our article. Your comments and suggestions were genuinely helpful. We are especially grateful for your thoughts and ideas on the introduction and discussion sections of the paper. We all agree that your thoughtful perspective and constructive feedback improved the article in ways we had not thought of before.
Point 1: Abstract: I suggest including the objective of the investigation.
Response 1: We reworded the first sentence of the abstract to clearly state the objective.
L25: “The objective of this study was to evaluate Freedom-Ranger broiler chickens as a method …”
Point 2: Introduction: In the introduction, reference is made to problems caused by species of the Diptera order to cattle. Among them are named and described the damage caused by two species: M. autumnalis and Haematobia irritans. Their life cycles are described, ending with the need to search for disruption methods to control these pests. However, the criteria for choosing to work with the M. autumnalis and not with H. irritans is not indicated. I suggest removing the information regarding H. irritans or including the criteria to choose only one species.
Response 2: Thank you for providing your perspective on how to make the introduction clearer. We removed information regarding H. irritans throughout the introduction.
Point 3: Results and discussion: Figure 2 is the most important in the study. With this figure, the authors discuss and conclude that broilers do not significantly affect M. autumnalis survival (HIGH, NEG and LOW) compared to GH treatment. Although the possible factors that could have influenced these results are discussed, there are important points that could have been discussed. For example, the authors in the methodology describe a wide botanical variety in the pastures on which the cows were fed. Among them, there is Lolium perenne which is a forage species characterized by presenting a symbiotic association with chemical compounds of the alkaloid type (peramine, lolitrem B among others). It has been determined that these compounds are transferred from the feeding of the animal through its digestive tract and are eliminated through its dung. In an interesting study, it was determined that these dung had traces of these alkaloids and significantly reduced the survival of H. irritans in laboratory and field conditions (see Parra, L., A. Mutis, M. Chacón, M. Lizama, C. Rojas, A. Catrileo, O. Rubilar, G. Tortella, M.A. Birkett & A. Quiroz. 2016. Horn fly larval survival in cattle dung is reduced by endophyte infection of tall fescue pasture. Pest Management Science 72(7): 1328-1334.). This fact could be part of the discussion in terms of explaining the low survival of larvae from the treatments since they were kept in dung that came from the field.
Response 3: Thank you for providing this information and reference!
We have added the following paragraph to the discussion section (L373):
“It is possible that the cattle dung use in this study reduced the survival of fly larvae due to the presence of alkaloids. A recent study by Parra et al. [36] reported that the dung from cattle that consumed endophyte-infected fescue contained two important alkaloids, peramine and lolitrem B, which reduced horn fly (H. irritans) larval survival from 45% to 10%. Based on this information, it is possible that the consumption of perennial ryegrass (L. perenne) included in the pastures of the current study resulted in the presence of these alkaloids and consequently lowered the survival of the fly larvae.”
Point 4: Lines 299-301. “Results previous research agrees with….” What are those results? References?
Response 4: Thank you for catching this statement that requires clarification! This statement is referring to the previously mentioned studies. We have changed the wording to clarify that.
L309: “… previous research [29–31] agrees with …”
Point 5: I have the feeling that this work focused more on the behavior of chickens in the face of climatic factors than on the effectiveness of broilers as controllers of face fly. Although the title incorporates both themes (efficacy and behavior), the research question of this work is whether or not there is a controlling effect of broilers on the face fly?
Response 5: You are correct that the main research question was whether or not there was a controlling effect of the broilers on the larvae survival. This research also focused on behavior to supplement the results of the main research question. The success of using broilers as a method of fly larvae control is contingent on broiler behaviors, which we know from previous studies can be quite variable depending on numerous factors (e.g., weather). We also decided to include an evaluation of behavior based on producer input; producers emphasized the importance of understanding poultry behavior in order for this method of fly larvae control to even be feasible. Much of the evidence for this method is anecdotal so gaining a deeper understanding of why broilers may or may not be successful in reducing larva survival was deemed as important for this study.
Reviewer 2 Report
General comments: The manuscript is well written and clearly structured. The introduction gives a thorough description of the background. The aim of the study is original and interesting in an organic perspective. I enjoyed reading the paper and only have minor comments, see below. One of may main concerns is that there are several influencing factors that may affect your results that you have not discussed: hybrid, leg health of the birds, larva palatable ++.
The title: the efficacy is highly influenced by the broilers behaviour, which in turn is influenced by climatic conditions. Therefore, I suggest you remove “and behaviors” from the title. This will also make the title easier to read.
Line 119: please include hybrid name. This is important since different hybrids may vary in their time budget.
Line 121: how were the birds housed prior to the experiment? Rearing conditions is likely to affect adult behavior.
Line 244-246: you describe the pasture as mostly grass and perennials and no trees. Is this right? If so, I would presume that the lack of trees or other cover from predator birds will highly affect the broilers pasture use. Trees would also give important shelter from radiation. Since the pasture was not designed from a poultry point of view, hence no threes, I suggest you rephrase the conclusion in line 244-246. As it is stated now is sounds like broilers are generally unwilling to use pasture and I do not believe this is correct if the pasture is created to suit the broiler’s needs.
Line 251: these results suggests that the hybrid used in this study under these conditions (no trees) reduces activity in the sun. Please modify the statement.
Line 288-295: in this section you refer to several studies using specific poultry species. This is yet another argument for why you need to state the hybrid used in your study. A broiler is not a broiler.
Line 299-200: “Results from previous studies…” Please insert references here. In addition, this would be a natural place to discuss different broiler hybrids utility and behavior.
Could the age of the birds affect the results? Should be discussed.
Line 350: given the same suboptimal pasture habitat, do you think a different climate would affect the pasture ranging?
What was the weight of the birds? Did you perform any lameness assessment, like gait scoring, prior to the study? If a large proportion of the birds had gait difficulties this would likely affect the results, since these broilers may be reluctant to move and spend a lot of time sitting. This is in accordance with the observation stated in line 369. Again, this may be highly influenced by hybrid, as well as weight and health. Should be discussed.
Do you know if the time budget of the birds in your study is in line with the time budget of the same type of broiler kept indoors in a cooler climate?
Line 375-376: please specify that your statement is only relevant for an uncovered pasture and the specific broiler hybrid used in this study.
Any thoughts on how foraging in cow dung may affect food security and prevalence of Salmonella in poultry meat?
Is the face fly larva attractive for broilers if they do not have to go outside or in the dung to get it? Please add some background information about this. Could the result be affected by the fact that the broilers don’t find this larva palatable?
Author Response
Thank you so much for taking the time to review our article. Your comments and suggestions were genuinely helpful. We are especially grateful for your thoughts and ideas on including the hybrid name throughout the paper and discussing topics related to results of this research. We all agree that your thoughtful perspective and constructive feedback improved the article in ways we had not thought of before.
Point 1: The title: the efficacy is highly influenced by the broiler’s behavior, which in turn is influenced by climatic conditions. Therefore, I suggest you remove “and behaviors” from the title. This will also make the title easier to read.
Response 1: We updated the title of the manuscript:
“Efficacy of broilers as a method of face fly (Musca autumnalis De Geer) larva control for organic dairy production”
Point 2: Line 119: please include hybrid name. This is important since different hybrids may vary in their time budget.
Response 2: The hybrid name of the broilers (Freedom Ranger) is given in line 114. We clarified that this is a four-line hybrid and included more information about the hybrid by including the following sentence (L115):
“The Freedom Ranger hybrid consisted of a four-line cross developed in the 1960’s to meet the French Label Rouge Free Range program standards.”
Point 3: Line 121: how were the birds housed prior to the experiment? Rearing conditions is likely to affect adult behavior.
Response 3: The following sentence was added (L117): “For each replicate group, day-old broiler chicks that were vaccinated for Marek’s at the hatchery were housed in a 2.22-m2 pen bedded with 5 cm of wood shavings and had ad libitum access to feed and water.”
Point 4: Line 244-246: you describe the pasture as mostly grass and perennials and no trees. Is this right? If so, I would presume that the lack of trees or other cover from predator birds will highly affect the broilers pasture use. Trees would also give important shelter from radiation. Since the pasture was not designed from a poultry point of view, hence no threes, I suggest you rephrase the conclusion in line 244-246. As it is stated now is sounds like broilers are generally unwilling to use pasture and I do not believe this is correct if the pasture is created to suit the broiler’s needs.
Response 4: Thank you for your input here! Yes, you are correct. The pasture space used in the study represented what is most commonly used in organic dairy cattle production in the upper Midwest, characterized by open pasture and no tree cover. The following statement has been included (L252):
“… for the experimental conditions of the study, which were characterized by open cattle grazing pasture and lack of tree cover.”
Point 5: Line 251: these results suggests that the hybrid used in this study under these conditions (no trees) reduces activity in the sun. Please modify the statement.
Response 5: This statement has been modified as requested (L258):
“These results suggest that the hybrid broilers used in this study under these experimental conditions characterized by no tree cover avoid overheating by reducing activity in sun and seek shade as levels of heat stress increase.”
Point 6: Line 288-295: in this section you refer to several studies using specific poultry species. This is yet another argument for why you need to state the hybrid used in your study. A broiler is not a broiler.
Response 6: The specific hybrid name (Freedom Ranger) is now specified when discussing broilers throughout the discussion section.
Point 7: Line 299-200: “Results from previous studies…” Please insert references here. In addition, this would be a natural place to discuss different broiler hybrids utility and behavior.
Response 7: The references were inserted as requested. The following paragraph was also included to discuss different broiler hybrids and their utility and behavior as requested (L320):
“The Freedom Ranger broilers used in the current study were a hybrid cross of several strains and were originally bred for their suitability for pasture housing systems. Previous studies [37,38] suggest that hybrid strains that demonstrate slower growth and improved mobility compared to pure strains make hybrids more equipped for free-range systems. A study by Lambertz et al. [37] demonstrated that hybrid male broilers (Bresse-Gauloise × New Hampshire) grew faster but had similar carcass quality to their purebred equivalent (Bresse-Gauloise). Other slow-growing hybrid broilers (Labresse × L86) were previously demonstrated to have increased pasture ranging, improved gait scores, and reduced dermal lesions compared to their fast-growing counterpart (Ross 208) [38]. Alternatively, another study [39] reported that a slow-growing pure strain (White Bresse L40) demonstrated increased pasture ranging and foraging behaviors compared to a fast-growing hybrid strain (Kosmos 8 Red), suggesting that speed of growth may be an important influencer of broiler behavior.”
Point 8: Could the age of the birds affect the results? Should be discussed.
Response 8: Thank you recommending this topic in the discussion as it is relevant for future studies. We have included the following paragraph (L420):
“It is unknown whether or not broilers of an age group older than that used in the current study (approximately 7-weeks-old) would have yielded different study results. Fanatico et al. [46] reported that outdoor foraging events increased by a factor of 1.7 between 7 and 10 week-old slow-growing broilers. Almeida et al. [39] similarly reported that outdoor foraging increased between the ages of 11 and 15 weeks for both slow- and fast-growing broiler strains, but acknowledged that broilers rarely consumed larvae or pupae based on an analysis of crop contents. Broilers of the current study were never observed foraging in dung pats for larvae and it seems reasonably unlikely that they would suddenly include this novel foraging technique to their behavioral repertoire as they approach slaughter weight at approximately 12 weeks of age. Yet, it is also possible for broilers to learn specialized foraging strategies depending on the social structure of the flock since domestic fowl engage in social learning during foraging events [51]. Therefore, an interaction between age and social learning of the flock may affect the success of foraging for fly larvae in dung pats for broilers.”
Point 9: Line 350: given the same suboptimal pasture habitat, do you think a different climate would affect the pasture ranging?
Response 9: Yes, it is likely that a more temperate climate with cooler summers would help promote pasture ranging given the same suboptimal pasture habitat commonly used for cattle grazing. However, cattle pastures without trees or shelters would still be unsuited for promoting pasture ranging since they do not provide protection from predators. To reiterate that pasture ranging is contingent on many important factors, two of which are protection from heat and protection from predators, the following statement was added (L393):
“It is possible that a more temperate climate with cooler summers would help promote pasture ranging in a similar treeless pasture habitat used for cattle grazing. However, cattle pastures without trees or shelters would still be unsuited for promoting pasture ranging even under optimal weather conditions since they do not provide protection from predators.”
Point 10: What was the weight of the birds? Did you perform any lameness assessment, like gait scoring, prior to the study? If a large proportion of the birds had gait difficulties this would likely affect the results, since these broilers may be reluctant to move and spend a lot of time sitting. This is in accordance with the observation stated in line 369. Again, this may be highly influenced by hybrid, as well as weight and health. Should be discussed.
Response 10: Thank you for your input here. Line 155 states that “… average broiler weight was 1.94 ± 0.7 kg.” The health of each bird was assessed prior the study, starting at 4 weeks of age and weekly thereafter. Body weight, feather (wing, tail, back, and thigh) scores, hock lesions, and foot pad lesions were assessed during each weekly heath assessment. No birds had hock or foot pad lesions and feather damage was minimal throughout the study. Although, we did not formally gait score each bird during these assessments, we did check all birds for abnormal gaits; no birds had obvious signs of gait difficulties and all birds could run (which was often observed during evening feeding events when the birds realized their caretaker had arrived to deliver their feed). The following statement was added (L133):
“The health of each bird was assessed prior to study initiation, starting at 4 weeks of age and weekly thereafter. Hock lesions and foot pad lesions were assessed during each weekly heath assessment. No broilers had hock or foot pad lesions or showed signs of gait difficulties.”
Point 11: Do you know if the time budget of the birds in your study is in line with the time budget of the same type of broiler kept indoors in a cooler climate?
Response 11: We could not find any behavioral studies on Freedom Ranger broilers housed indoors that reported a time budget. However, we found one study that reported a time budget for Rowan Rangers (another slow-growing meat-type broiler) reared indoors (Wallenbeck et al., 2016).
This table compares the time budgets for Wallenbeck et al. to the current study, where only three similar behaviors are reported:
|
Behavior |
Wallenbeck et al., 2016 |
Current study |
|
Sitting |
57% |
41% |
|
Standing |
36% |
30% |
|
Sleeping |
8% |
14% |
Wallenbeck, A.; Wilhelmsson, S.; Jönsson, L.; Gunnarsson, S.; Yngvesson, J. Behaviour in one fast-growing and one slower-growing broiler (Gallus gallus domesticus) hybrid fed a high- or low-protein diet during a 10-week rearing period. Acta Agric. Scand. Sect. A — Anim. Sci. 2016, 66, 168–176.
Point 12: Line 375-376: please specify that your statement is only relevant for an uncovered pasture and the specific broiler hybrid used in this study.
Response 12: We have modified the statement as requested (L432):
“To our knowledge, this is the first study to provide evidence that Freedom Ranger broilers do not forage for face fly larvae in cow dung pats in uncovered cattle pasture.”
Point 13: Any thoughts on how foraging in cow dung may affect food security and prevalence of Salmonella in poultry meat?
Response 13: Great question. I do not think this is a well understood topic but is becoming increasingly important, especially considering systems where multiple livestock species share the same space. I do think there could be a risk for food security issues in terms of Salmonella prevalence in poultry meat when broilers forage in cow dung, which would be contingent on the prevalence of Salmonella in cattle dung. Fossler et al. (2004) investigated the prevalence of Salmonella in organic dairy cattle feces on 26 farms in Minnesota, Wisconsin, Michigan, and New York and found that Salmonella was present in 4.7% of the 4,353 feces samples taken every 2 months over a 1-year period. Nazareth et al. (2019) also reported that Salmonella was isolated from 3.3% of fecal samples from grass-fed dairy steers in Iowa, Minnesota, and Pennsylvania.
I also think that the risk for Salmonella contamination of pasture-raised broiler meat is multifaceted and that other risk factors, such as transmission from wildlife and butchering methods, should be considered. A study (Pires et al., 2019) performed on 20 diversified livestock farms in California where multiple livestock species were raised together found that none of the 46 swabs of broiler house drags, broiler house surfaces, or broiler feces were positive for Salmonella, suggesting that shared housing between multiple species may not be a major risk for Salmonella contamination in broilers (yet, this study was small and more studies are need to confirm this idea). Another study (Hwang et al., 2020) reported that the prevalence of Salmonella in 433 soil samples from broiler pastures in the summer on 11 farms in southeastern United States was 11%, suggesting that Salmonella may be naturally present in the pasture environment.
Although this is not within the scope of this research, for diversified livestock systems, the health of other livestock species and local wildlife sharing the same space as broilers should be monitored and managed in a manner that reduces the risk of Salmonella prevalence in poultry meat.
Pires, A.F.A.; Patterson, L.; Kukielka, E.A.; Aminabadi, P.; Navarro-Gonzalez, N.; Jay-Russell, M.T. Prevalence and risk factors associated with Campylobacter spp. and Salmonella enterica in livestock raised on diversified small-scale farms in California. Epidemiol. Infect. 2019, 147.
Fossler, C.P.; Wells, S.J.; Kaneene, J.B.; Ruegg, P.L.; Warnick, L.D.; Bender, J.B.; Godden, S.M.; Halbert, L.W.; Campbell, A.M.; Zwald, A.M.G. Prevalence of Salmonella spp on conventional and organic dairy farms. J. Am. Vet. Med. Assoc. 2004, 225, 567–573.
Nazareth, J.; Shaw, A.; Delate, K.; Turnbull, R. Food safety considerations in integrated organic crop-livestock systems: Prevalence of Salmonella spp. and E. coli O157:H7 in organically raised cattle and organic feed. Renew. Agric. Food Syst. 2019, 1–9.
Hwang, D.; Rothrock, M.J.; Pang, H.; Guo, M.; Mishra, A. Predicting Salmonella prevalence associated with meteorological factors in pastured poultry farms in southeastern United States. Sci. Total Environ. 2020, 713.
Point 14: Is the face fly larva attractive for broilers if they do not have to go outside or in the dung to get it? Please add some background information about this. Could the result be affected by the fact that the broilers don’t find this larva palatable?
Response 14: Great point! At the end of the study, we actually tested to see if the broilers would eat the larvae and pupae we collected. Indeed, they ate it when it was presented in our hands and on the soil. An early study (Dashefsky et al., 1976) successfully fed face fly pupae to chickens, yet level of palpability was not determined. Other studies (Schiavone et al., 2018; Hwangbo et al., 2009; Wang and Shelomi, 2017; Ramos-Elorduy et al., 2002) suggest that black soldier fly, house fly, and yellow mealworm larvae can successfully be used as a palatable feed supplement for poultry. Although research on the palatability of face fly larvae for poultry is sparse, other insect larvae was previously demonstrated to be highly palatable for poultry.
The following paragraph has been included (L312):
“It is unknown whether face fly larvae are truly an attractive feed source for broilers. Anecdotal evidence suggests that face fly larvae and pupae are highly sought after when they are hand-fed to broilers. In an early study, Dashefsky et al. [32] successfully fed face fly pupae to day-old White Rock chickens, yet the level of palpability was not determined or discussed. Other studies [33–36] suggested that black soldier fly (Hermethia illucens L.), house fly (Musca domestica L.), and yellow mealworm (Tenebrio molitor L.) larvae can successfully be used as a palatable feed supplement for poultry. Although research on the palatability of face fly larvae for poultry is sparse, insect larvae in general is a highly attractive feed source for poultry.”
Schiavone, A.; Dabbou, S.; De Marco, M.; Cullere, M.; Biasato, I.; Biasibetti, E.; Capucchio, M.T.; Bergagna, S.; Dezzutto, D.; Meneguz, M.; et al. Black soldier fly larva fat inclusion in finisher broiler chicken diet as an alternative fat source. Animal 2018, 12, 2032–2039.
Hwang, D.; Rothrock, M.J.; Pang, H.; Guo, M.; Mishra, A. Predicting Salmonella prevalence associated with meteorological factors in pastured poultry farms in southeastern United States. Sci. Total Environ. 2020, 713, 136359.
Wang, Y.-S.; Shelomi, M. Review of black soldier fly (Hermetia illucens) as animal feed and human food. Foods 2017, 6, 91.
Ramos-Elorduy, J.; González, E.A.; Hernández, A.R.; Pino, J.M. Use of Tenebrio molitor (Coleoptera: Tenebrionidae) to recycle organic wastes and as feed for broiler chickens. J. Econ. Entomol. 2002, 95, 214–220.
Dashefsky, H.S.; Anderson, D.L.; Tobin, E.N.; Peters, T.M. Face fly pupae: A potential feed supplement for poultry. Environ. Entomol. 1976, 5, 680–682.
Reviewer 3 Report
The objective of this study was to assess whether free-range broilers can be a pest-control management strategy for face fly larva in organic dairy production system. There is a lack of empirical data on pest control management using poultry in cattle production systems, and this manuscript add relevant information for the development of effective pest-control management practices in organic production systems. The results of this research suggest that free-range finisher broilers were ineffective at lowering the growth rate of larva in cow dung. However, I personally think that the discussion is missing a critical analysis of the reasons behind of the lack of differences between control and both broiler groups, particularly looking at poultry behaviour. The proportion of time that broilers spent ranging in this study is very low compared to time-budget standards for chickens (e.g. Dawkins, 1989), probably due to strain differences (finisher meat chickens vs layers and dual-purposed chickens) and at a market body weight. Chickens spend most of the time foraging at dawn and dusk, and the methodology used in this study to describe the foraging behaviour of free-range broiler may not have been sensitive to capture these two daily peaks in foraging activity. Additionally, the development of efficient foraging behaviour is a learning process that requires experience. In this case, chickens need to learn the nutritional values of fly larva and find them in cow dungs. Under natural settings, hens will teach chicks how forage and where to find food. Under commercial conditions, chicks require previous experience and trail-and-error. Poultry species often scratch, seek for food, and forage on manure; but in this study, broilers lacked time and previous experience with cow dungs to learn the nutritional value of larvae. For this reason, experienced (free-range layers and dual-purposed chickens) vs naïve poultry (finisher broilers) may be a more feasible approach to control the larva growth rate in cow dungs. I think that including these ideas in the discussion will enhance the quality of the paper, the interpretation of the results, and generate hypotheses for future research.
The manuscript is clear and well written. I include below few minor comments for clarity in the Methodology and Results’ sections.
LINE 119: Include further details about their rearing: vaccinations, housing, and management during the first weeks of life. Were chicks exposed to forage/enrichment before placement on pasture?
LINE 131: add “per bird” after “of concentrate”.
LINE 154: Add at the end of the sentence: at the onset of the experiment. Or similar.
LINE 161: Insert subheading: Data collection.
LINE 164: The behavioural sampling method needs further detail. How many one-min observation periods per bird and day? How did you control for time of day and pen/treatment during the behavioural sampling?
LINE 168: Binary outcomes in confusing. Does it mean that the occurrence of foraging behaviours within the one-min observation period was recorded as yes (presence) or no (absence)? What about if the behaviour was performed more than once (multiple bouts of foraging) within the focal observation period? I suggest clarifying this along the manuscript (including in larval survival stages).
Table 1. I suggest replacing travel by locomotion.
LINE 221: Were there signs of scratching or pecking behaviour?
LINE 251-252: This sentence belongs to the discussion section.
Figure 3. I don’t find easy to difference between the standing and locomotion trendline. I suggest choosing a more visually friendly legend.
LINE 268-271: I suggest rewording both sentences because It might be misleading due to the proportion of time standing and sleeping is also percentage. For example: From 20 to 35 C (or whatever sampling temperature threshold applies to these data), the proportion of time broiler spent standing increased 1.1 times for every one degree increase in CCI.
Author Response
Thank you so much for taking the time to review our article. Your comments and suggestions were genuinely helpful. We are especially grateful for your thoughts and ideas on including a discussion about the development of foraging behaviors in poultry. We all agree that your thoughtful perspective and constructive feedback improved the article in ways we had not thought of before.
Point 1: I personally think that the discussion is missing a critical analysis of the reasons behind of the lack of differences between control and both broiler groups, particularly looking at poultry behaviour. The proportion of time that broilers spent ranging in this study is very low compared to time-budget standards for chickens (e.g. Dawkins, 1989), probably due to strain differences (finisher meat chickens vs layers and dual-purposed chickens) and at a market body weight. Chickens spend most of the time foraging at dawn and dusk, and the methodology used in this study to describe the foraging behaviour of freerange broiler may not have been sensitive to capture these two daily peaks in foraging activity. Additionally, the development of efficient foraging behaviour is a learning process that requires experience. In this case, chickens need to learn the nutritional values of fly larva and find them in cow dungs. Under natural settings, hens will teach chicks how forage and where to find food. Under commercial conditions, chicks require previous experience and trail-and-error. Poultry species often scratch, seek for food, and forage on manure; but in this study, broilers lacked time and previous experience with cow dungs to learn the nutritional value of larvae. For this reason, experienced (freerange layers and dual-purposed chickens) vs naïve poultry (finisher broilers) may be a more feasible approach to control the larva growth rate in cow dungs. I think that including these ideas in the discussion will enhance the quality of the paper, the interpretation of the results, and generate hypotheses for future research.
Response 1: Thank you so much for providing your insight on these topics to include in the discussion. We also agree that the discussion section would be enhanced by identifying the possible limitation of the methodology used to estimate foraging behavior and discussing the development of efficient foraging behavior in poultry. These are very important topics to consider for future studies.
The following was added to the discussion section (L417): “A possible limitation of this study includes infrequent behavioral sampling. Continuous sampling from dusk until dawn would provide a more accurate estimate of behaviors.”
The following was added to the discussion section (L420):
“It is unknown whether or not broilers of an age group older than that used in the current study (approximately 7-weeks-old) would have yielded different study results. Fanatico et al. [46] reported that outdoor foraging events increased by a factor of 1.7 between 7 and 10 week-old slow-growing broilers. Almeida et al. [39] similarly reported that outdoor foraging increased between the ages of 11 and 15 weeks for both slow- and fast-growing broiler strains, but acknowledged that broilers rarely consumed larvae or pupae based on an analysis of crop contents. Broilers of the current study were never observed foraging in dung pats for larvae and it seems reasonably unlikely that they would suddenly include this novel foraging technique to their behavioral repertoire as they approach slaughter weight at approximately 12 weeks of age. Yet, it is also possible for broilers to learn specialized foraging strategies depending on the social structure of the flock since domestic fowl engage in social learning during foraging events [51]. Therefore, an interaction between age and social learning of the flock may affect the success of foraging for fly larvae in dung pats for broilers.”
Point 2: LINE 119: Include further details about their rearing: vaccinations, housing, and management during the first weeks of life. Were chicks exposed to forage/enrichment before placement on pasture?
Response 2: The following sentence was added (L117): “For each replicate group, day-old broiler chicks that were vaccinated for Marek’s at the hatchery were housed in a 2.22-m2 pen bedded with 5 cm of wood shavings and had ad libitum access to feed and water.”
Point 3: LINE 131: add “per bird” after “of concentrate”.
Response 3: This statement was modified as requested (L130)
Point 4: LINE 154: Add at the end of the sentence: at the onset of the experiment. Or similar.
Response 4: This statement was modified as requested (L156)
Point 5: LINE 161: Insert subheading: Data collection.
Response 5: A subheading was inserted as requested (L157)
Point 6: LINE 164: The behavioural sampling method needs further detail. How many one-min observation periods per bird and day? How did you control for time of day and pen/treatment during the behavioural sampling?
Response 6: There were two 60-s observation periods per focal broiler per day corresponding to the morning and afternoon observations. Observations occurred within the sampling windows (L165) to account for time of day and focal birds were observed in random order alternating between treatment pens to control for pen/treatment.
The following was included in L166: “…for a total of two observation periods per day.”
The following was included in L170: “… and were observed in random order alternating between treatment pens.”
Point 7: LINE 168: Binary outcomes in confusing. Does it mean that the occurrence of foraging behaviours within the one-min observation period was recorded as yes (presence) or no (absence)? What about if the behaviour was performed more than once (multiple bouts of foraging) within the focal observation period? I suggest clarifying this along the manuscript (including in larval survival stages).
Response 7: Yes, you are correct; the outcome for foraging was recorded as either a yes or no. If multiple foraging bouts were observed, they were still recorded as “yes”.
The following was included in L172: “… (i.e., the occurrence of foraging behaviors within the 60-s observation period was recorded as either a yes [presence] or no [absence]). The frequency of foraging bouts was not recorded.”
Point 8: Table 1. I suggest replacing travel by locomotion.
Response 8: We decided to not change the wording in the table in order to keep consistency throughout the paper.
Point 9: LINE 221: Were there signs of scratching or pecking behaviour?
Response 9: There were no indication of foraging in the pats.
The following was added (L227): “…but not scratching nor pecking…”
Point 10: LINE 251-252: This sentence belongs to the discussion section.
Response 10: This sentence was included to translated numerical results to lay terms. This sentence does not provide any further discussion of the results reported in this paragraph, so we decided it was necessary to include in the results section of the paper rather than the discussion section.
Point 11: Figure 3. I don’t find easy to difference between the standing and locomotion trendline. I suggest choosing a more visually friendly legend.
Response 11: Thank you for catching this! We made the lines thinner in order to differentiate between line types.
Point 12: LINE 268-271: I suggest rewording both sentences because It might be misleading due to the proportion of time standing and sleeping is also percentage. For example: From 20 to 35 C (or whatever sampling temperature threshold applies to these data), the proportion of time broiler spent standing increased 1.1 times for every one degree increase in CCI.
Response 12: Thank you for your input on this. The following statement has been modified (L277): “… increased by a factor of 1.10 (regression coefficient = 0.10, 95% CI = 0.02–0.17). Alternatively, the proportion of time broilers were observed sleeping decreased by a factor of 0.92…”